# Synaptic Targets of Glycinergic Neurons in Laminae I–III of the Spinal Dorsal Horn

**DOI:** 10.3390/ijms24086943

**Published:** 2023-04-08

**Authors:** Camila Oliveira Miranda, Krisztina Hegedüs, Gréta Kis, Miklós Antal

**Affiliations:** Department of Anatomy, Histology and Embryology, Faculty of Medicine, University of Debrecen, H-4032 Debrecen, Hungary

**Keywords:** glycine, spinal dorsal horn, pain processing neural circuits, transgenic animals, immunocytochemistry, in situ hybridization, electron microscopy

## Abstract

A great deal of evidence supports the inevitable importance of spinal glycinergic inhibition in the development of chronic pain conditions. However, it remains unclear how glycinergic neurons contribute to the formation of spinal neural circuits underlying pain-related information processing. Thus, we intended to explore the synaptic targets of spinal glycinergic neurons in the pain processing region (laminae I–III) of the spinal dorsal horn by combining transgenic technology with immunocytochemistry and in situ hybridization accompanied by light and electron microscopy. First, our results suggest that, in addition to neurons in laminae I–III, glycinergic neurons with cell bodies in lamina IV may contribute substantially to spinal pain processing. On the one hand, we show that glycine transporter 2 immunostained glycinergic axon terminals target almost all types of excitatory and inhibitory interneurons identified by their neuronal markers in laminae I–III. Thus, glycinergic postsynaptic inhibition, including glycinergic inhibition of inhibitory interneurons, must be a common functional mechanism of spinal pain processing. On the other hand, our results demonstrate that glycine transporter 2 containing axon terminals target only specific subsets of axon terminals in laminae I–III, including nonpeptidergic nociceptive C fibers binding IB4 and nonnociceptive myelinated A fibers immunoreactive for type 1 vesicular glutamate transporter, indicating that glycinergic presynaptic inhibition may be important for targeting functionally specific subpopulations of primary afferent inputs.

## 1. Introduction

In addition to gamma amino butyric acid (GABA), which is the dominant fast inhibitory neurotransmitter in the central nervous system, glycine-mediated inhibition plays essential roles in spinal cord functions, including nociceptive and pain-related information processing in the superficial layers (laminae I–III) of the dorsal horn [1,2,3]. In the 1990s, immunohistochemical observations indicated strong colocalization between GABA and glycine in laminae I–III of the spinal dorsal horn: specifically, all glycinergic neurons contain GABA, but there is a substantial proportion of GABAergic neurons in which the neurotransmitter glycine cannot be detected [4,5,6,7,8]. However, in addition to mixed GABA-glycine and GABA-only mIPSCs, glycine-only mIPSCs have been frequently recorded in neurons in the superficial spinal dorsal horn [9,10]. Furthermore, almost all physiological and pharmacological experimental data from the last two decades have substantiated the importance of glycine-only inhibition in the spinal dorsal horn, and its fundamental role in pain-related information processing [1,11,12,13,14,15,16]. There is compelling evidence that reduced glycinergic synaptic transmission induces central sensitization, hyperalgesia, and allodynia, leading to the development of chronic pain conditions [1,12,17,18,19,20,21,22]. Strongly reinforcing the importance of glycinergic inhibition, by using a combination of electrophysiology and optogenetics to quantify changes in inhibitory synaptic transmission onto neurons in the superficial spinal dorsal horn caused by the ablation of glycinergic neurons, Foster et al. [1] found that more than 70% of the total inhibitory postsynaptic current amplitude recorded in excitatory neurons was blocked by the application of the glycine receptor antagonist strychnine and only the rest could be blocked by the GABA_A_ receptor antagonist bicuculline. In addition, a series of studies in which inhibitory quantal postsynaptic currents were recorded in adult neurons within the superficial spinal dorsal horn showed that postsynaptic currents mediated by either glycine or GABA_A_ receptors were common, but the incidence of mixed postsynaptic currents was much lower [16,23,24,25].

While there is a vast amount of physiological and pharmacological evidence emphasizing the importance of GABA-independent glycinergic inhibition, the first morphological data showing glycine-only axon terminals in the superficial spinal dorsal horn were reported recently [26]. First, Miranda et al. [26] provided experimental evidence indicating that there are glycine-only axon terminals in laminae I–III that do not express GABA. This was a substantial step forward in narrowing the gap between morphological and physiological findings, but the question of how glycinergic neurons contribute to the formation of spinal neural circuits underlying nociceptive and pain-related information processing remains unanswered. As a first step to answering this question, we intended to explore the synaptic targets of glycinergic axon terminals in laminae I–III of the spinal dorsal horn by combining transgenic technology with immunocytochemistry and in situ hybridization accompanied by light and electron microscopy.

## 2. Results

### 2.1. Elimination of GlyT2 Expressing Axon Terminals Descending from Higher Brain Centers to the Lumbar Spinal Cord

In the present work, we explored the synaptic relations of glycinergic axon terminals of spinal origin. However, in addition to spinal neurons, glycinergic axons descending from brain stem nuclei also make a substantial contribution to the glycinergic innervation of the spinal dorsal horn [27,28]. To eliminate descending glycinergic inputs, we hemisected the spinal cord at the Th11–Th12 level (Figure 1) and regarded the noneliminated glycinergic axon terminals in the lumbar spinal cord ipsilateral to the hemisection as terminals of spinal origin. Thus, GlyT2 immunostained axon terminals on the ipsilateral side were counted in sections of the L4–L5 spinal cord segments of animals subjected to hemisection, and this number was compared to the number of GlyT2 positive axon terminals in the corresponding segments in nonoperated animals 715 (Figure 2). In laminae I–II, we counted in total 230 and 156 (15.2 ± 3.3 and 10.4 ± 1.9 in the individual sections; *p* < 0.001) immunostained axon terminals in the nonoperated animals and animals subjected to hemisection, respectively. In lamina III, in total 715 and 432 (54.0 ± 5.3 and 28.9 ± 3.4 in the individual sections; *p* < 0.001) GlyT2 positive axon terminals were found in the nonoperated animals and animals that underwent hemisection, respectively. These findings indicate that at the L4–L5 level, approximately 68% and 60% of all GlyT2 immunoreactive axon terminals were of spinal origin in laminae I–II and lamina III, respectively, and the rest may have represented axon terminals of brainstem origin. These results also show that to study glycinergic axon terminals of spinal origin in the superficial lumbar spinal dorsal horn, a hemisection of the spinal cord at a low thoracic level hemisection is required. For this reason, the following experiments were performed exclusively on animals that underwent hemisection.

### 2.2. Synapses Formed by GlyT2 Immunoreactive Axon Terminals

The NiDAB-silver-gold detection method revealed the presence of GlyT2-containing axon terminals with high accuracy. The staining was remarkably intense, with minimal, if any, background. Gold particles representing GlyT2-like immunoreactivity were observed in contact with the intracellular side of the extrasynaptic cell membrane and over synaptic vesicles (Figure 3 and Figure 4), confirming earlier reports by Spike et al. [29] and Nunez et al. [30]. Occasionally, we also found some gold particles over mitochondria (Figure 3B,C), but this was probably nonspecific labeling due to the diffusion of the DAB reaction end-product within the labeled axon terminal. The size and shape of the immunoreactive boutons varied widely, but most of them were in the range of 0.5–1.0 µm. The labeled boutons formed symmetric synaptic contacts and contained flattened and/or rounded clear vesicles, but they were free of dense-core vesicles. We observed 138 labeled boutons, of which 104 (75.4%) formed axo-dendritic (Figure 3A–D), 16 (11.6%) axosomatic (Figure 3E,G), and 18 (13.0%) axo-axonic synaptic contacts (Figure 3F and Figure 4). Of the 18 labeled boutons that formed axo-axonic contacts, 7 participated in nonglomerular (Figure 3F) and 11 in glomerular (Figure 4) synaptic arrangements. The sizes of the postsynaptic dendrites and axons in nonglomerular synaptic arrangements were similar to the size of the GlyT2 immunostained boutons. The postsynaptic axons in the nonglomerular axo-axonic arrangements contained spherical vesicles and mitochondria but were free of any dense-core vesicles (Figure 3F). The large central axons of the synaptic glomeruli, to which immunostained boutons were associated, had electrolucent cytoplasm and contained many mitochondria as well as clusters of spheroid clear synaptic vesicles (Figure 4), and thus, they could be identified as type II synaptic glomeruli [31,32]. In addition to the GlyT2 immunoreactive axons, the central axon terminals of the synaptic glomeruli also made contact with other nonimmunoreactive axons and formed asymmetric synapses with dendrites (Figure 4).

### 2.3. Neurochemical Identification of Neurons Receiving Axosomatic Glycinergic Inputs

Electron microscopy showed that most of the glycinergic inputs of spinal origin contacted the somatodendritic compartment of neurons in laminae I–III of the spinal dorsal horn. To explore the types of interneurons that are targets of glycinergic inhibition we performed double immunostaining for GlyT2 and various cellular markers including CaB, PKCγ, CR, GAL, nNOS, and PV, which have been previously reported as markers of inhibitory and excitatory neurons in laminae I–III of the spinal dorsal horn [33,34,35,36,37,38,39,40]. Neurons that can be identified with these cellular markers have been reported to be heterogeneous in the sense that one single cellular marker can label both excitatory and inhibitory neurons [41,42,43]. Because we also wanted to explore whether glycinergic innervation is received by inhibitory and/or excitatory neurons, we performed the double immunostaining on sections obtained from Pax2:Cre-tdTomato transgenic animals. In these animals, neurons positive for PAX2, a transcription factor that is regarded as the most reliable marker of inhibitory neurons in the central nervous system, also expressed tdTomato [44,45,46].

Before performing triple labeling (PAX2-GlyT2-cellular marker) we wanted to check the specificity of tdTomato expression in the Pax2:Cre-tdTomato transgenic mice. Thus, by using multiple fluorescence in situ hybridization (FISH), we investigated the colocalization of tdTomato mRNA with GAD65/67 and GlyT2 mRNAs, markers that can label all inhibitory neurons in the superficial spinal dorsal horn. We followed the pretreatment and detection protocols provided by Advanced Cell Diagnostics and found that the probes for the mRNAs of interest were highly specific (Figure 5). Measuring the intensity of the background staining in laminae I–III, we found that there were 2.4 ± 1.4, 2.6 ± 1.5, and 8.2 ± 2.3 dots positive for GlyT2, GAD65/67, and tdTomato mRNAs, respectively, in 20 µm × 20 µm (400 µm^2^) cell-free squares. We regarded cellular signals as positive if they were at least three times stronger than the background signal calculated proportionally to the surface areas of the labeled neurons.

According to the above-described criteria, we observed 408 neurons positive for tdTomato mRNA in laminae I–III, and only 19 (4.6%) of these neurons were negative for both GlyT2 and GAD 65/67 mRNA (Figure 5J, Table 1). In addition, we found 9 neurons that expressed GlyT2 and/or GAD65/67 mRNA but were not labeled for tdTomato mRNA. The results showed that in the Pax2:Cre-tdTomato transgenic mice, tdTomato was highly expressed in inhibitory neurons and expressed in nearly all inhibitory neurons. We further studied the 389 neurons that were positive for tdTomato mRNA and also expressed GlyT2 and/or GAD 65/67 mRNA and found that in total 261 (67.1%) (17.4 ± 1.9 in the individual sections) expressed both GlyT2 and GAD65/67 mRNA, whereas, in total, 128 (32.9%) (8.6 ± 1.3 in the individual sections) showed positivity only for GAD65/67 mRNA (Figure 5A,C–E,G, Table 1). In neurons expressing both GlyT2 and GAD65/67 mRNA, the intensity of the signals was highly variable, from weak to strong, along a continuous scale, and the intensity of the signals for GlyT2 and GAD65/67 changed independently. Thus, we found cells with intense GAD65/67 mRNA labeling and weak/moderate GlyT2 mRNA labeling (Figure 5C,D), but there were also neurons in which GlyT2 mRNA labeling was stronger than GAD 65/67 mRNA labeling (Figure 5D,G). However, we did not observe any cells that were positive for GlyT2 mRNA and negative for GAD65/67 mRNA.

Although the number of neurons with strong GlyT2 expression was low, and neurons expressing only GLyT2 mRNA could not be detected in laminae I–III, it seems to be well-established that there are abundant glycine-only axon terminals in laminae I–III [26,47]. Because of the low number of neurons with strong GlyT2 mRNA labeling in laminae I–III, these glycine-only axon terminals in laminae I–III may arise from glycinergic neurons located in deeper laminae, especially lamina IV. To validate this assumption, we carefully investigated GlyT2 and GAD 65/67 mRNA expression in lamina IV. The background FISH signal was a lightly higher in this lamina, with 6.4 ± 2.1, 5.0 ± 1.9, and 14.2 ± 2.7 dots positive for GlyT2, GAD 65/67 and tdTomato mRNA, respectively, in cell-free 400 µm^2^ areas, but the labeling was also highly specific in this lamina (Figure 5B). Only 9 (1.4%) of the 665 tdTomato mRNA labeled neurons were negative for both GlyT2 and GAD 65/67 mRNAs (Table 1). Although GlyT2 mRNA labeling dominated in lamina IV neurons (Figure 5B,F,H,I), in total only 24 (3.7%) (1.6 ± 0.8 in the individual sections) of the 656 cells positive for GLYT2 and/or GAD 65/67 mRNAs expressed only GlyT2 mRNA (Figure 5I, Table 1). The proportion of these neurons was even lower than that of neurons that expressed only GAD 65/67 mRNA (in total 43, 6.5%; 2.9 ± 1.2 in the individual sections; Table 2). Similar to that was observed in laminae I–III, in neurons expressing both GlyT2 and GAD65/67 mRNA (in total 589, 89.8%; 39.2 ± 3.9 in individual sections; Table 1), the intensity of the signals was highly variable, from weak to strong, along a continuous scale, and the intensity of the signals for GlyT2 and GAD65/67 changed independently. However, unlike what was observed in laminae I–III, GlyT2 mRNA dominated in most of the cells in lamina IV (Figure 5B,F,H), whereas GAD65/67 mRNA staining was, with some exceptions, usually weak or moderate in these neurons (Figure 5F,H).

After showing that the Pax2:Cre-tdTomato transgene was highly specifically expressed in inhibitory neurons in laminae I–III, we performed triple immunolabeling of sections accordingly. First, we found that the CaB and CR immunoreactive neurons highly outnumbered the other populations of interneurons that we investigated. In the 15 samples (see the Materials and Methods for the sampling procedure), we found 356 and 346 neurons immunostained for CaB and CR, respectively (Table 2). PKCγ and PV-containing neurons also represented substantially large cell populations; 242 and 192 neurons were found to be immunostained for these markers (Table 2). However, in the same number of samples, we found only 62 and 21 neurons that were immunostained for nNOS and GAL, respectively (Table 2). Analysis of the colocalization of tdTomato with the cellular markers showed that only 3.1% of the CaB-containing neurons expressed tdTomato; thus, they could be regarded as excitatory neurons with high reliability (Table 2). Most PKCγ (86.8%) and CR (82.1%) immunoreactive neurons were negative for the PAX2:Cre-tdTomato transgene, but the rest of them expressed tdTomato (Table 2). Thus, although most PKCγ and CR-expressing neurons are excitatory, there are also a substantial number of inhibitory neurons expressing these markers. On the other hand, 90.5% of GAL-expressing neurons were tdTomato positive and thus inhibitory. While it seems that most CaB, PKCγ, and CR positive neurons are excitatory and most GAL-positive neurons are inhibitory, the inhibitory or excitatory nature of nNOS and PV neurons cannot be determined based only on the expression of the neuronal markers. Approximately half, more precisely 58.1% and 45.3%, of the nNOS and PV immunostained neurons, respectively, were positively labeled for tdTomato, whereas the rest were negative for PAX2:Cre-tdTomato. These results are in fundamental agreement with earlier reports [41,42,48,49,50,51,52,53,54], strongly supporting the reliability of our subsequent immunohistochemical data.

Following the above analysis, we intended to identify the population of interneurons that established close axosomatic appositions with axonal terminals immunoreactive for GlyT2. Our fundamental finding was that GlyT2 immunostained axon terminals made close axosomatic appositions with almost all groups of interneurons that we investigated (Figure 6 and Figure 7). We found numerous axosomatic contacts on CaB, CR, nNOS, and PV immunoreactive excitatory neurons not expressing tdTomato (Figure 6). There were few close appositions between GlyT2 immunostained axon terminals and the somata of PKCγ immunostained neurons not expressing tdTomato; such contacts were found primarily on proximal dendrites in the case of these neurons (Figure 6E). We did not observe close appositions between GlyT2 immunostained axons and the somata of excitatory GAL-containing neurons not expressing tdTomato. However, because we found only two GAL-positive neurons not expressing tdTomato in our sample, our finding was not necessarily reliable. In addition to excitatory neurons, GlyT2 immunostained axon terminals also came into contact with the somata of tdTomato-labeled PKCγ, CR, GAL, nNOS, and PV immunoreactive inhibitory neurons (Figure 7). We also revealed 11 CaB-containing neurons with tdTomato labeling, but the somata of these neurons were not in close appositions to GlyT2 immunostained axons. These results suggest that glycinergic postsynaptic inhibition can influence the activity of almost all populations of excitatory and inhibitory interneurons in laminae I–III of the spinal dorsal horn, thus it may have substantial and variable effects on spinal pain processing.

### 2.4. Neurochemical Identification of Axon Terminals Receiving Glycinergic Axo-Axonic Inputs

In addition to axosomatic and axo-dendritic synapses, electron microscopy showed that 13.0% of GlyT2 immunoreactive axon terminals formed axo-axonic synapses. To identify the types of axon terminals that can be the targets of glycinergic presynaptic inhibition, we combined GlyT2 immunostaining with IB4-binding and immunolabeling for CGRP, VGLUT1, and VGLUT2, markers of nonpeptidergic nociceptive afferents, peptidergic nociceptive afferents, nonnociceptive primary afferents, and the axon terminals of intrinsic excitatory spinal neurons, respectively, in laminae I–III of the spinal dorsal horn [55,56,57,58,59,60].

Based on a thorough investigation of 1 µm thick confocal sections, we found that GlyT2 immunoreactive axons did not form axo-axonic contacts with CGRP- or VGLUT2-expressing axons. However, we revealed close appositions between GlyT2 and VGLUT1 immunoreactive as well as IB4-binding axon terminals (Figure 8 and Figure 9). As indicated by the distribution of the postsynaptic axon terminals, all axo-axonic close appositions were exposed in inner lamina II and lamina III, whereas it seems that GlyT2 immunoreactive axon terminals did not form axo-axonic contacts in lamina I and outer lamina II.

### 2.5. Distribution of GlyT2 Immunostained Axon Terminals on the Somato-Dendritic Compartment of PKCγ-Expressing Neurons

Although we identified the types of neurons that receive glycinergic inputs by exploring axosomatic contacts, it is obvious that most of the inhibitory inputs are received by the dendrites of neurons. Thus, we were looking for a population of neurons that may serve as a model for analyzing the distribution of axo-dendritic glycinergic synapses along the dendritic tree. In addition, we assessed the dendritic location of synapses made by axon terminals of primary afferents, the function of which can be modulated by presynaptic glycinergic inhibition. For this reason, we selected the PKCγ-containing neurons which are known to receive substantial excitatory inputs from primary afferents and are under robust glycinergic pre- and postsynaptic inhibition [12,16,19,22,36,61,62,63,64,65]. In addition, their dendritic trees arborize in lamina II and lamina III [35,49,62,66,67], where we identified axo-axonic contacts between GlyT2 immunoreactive and IB4-binding as well as VGLUT1 immunoreactive axon terminals. We also took into consideration that 13.2% of PKCγ containing interneurons were found to be inhibitory, and inhibitory neurons can be under different glycinergic inhibitory control than excitatory neurons. Thus, we performed double and triple immunostaining of sections obtained from Prkcgtm2/cre/ERT2-tdTomato transgenic mice for GlyT2, PAX2, and IB4-binding or VGLUT1. The sections were divided into two groups and stained with an appropriate combination of antibodies: (1) anti-GlyT2 + IB4-biotin+ anti-PAX2, and (2) anti-GlyT2 + anti-VGLUT1 + anti-PAX2.

We identified 106 tdTomato-labeled PKCγ-containing neurons that received close appositions from IB4-binding or VGLUT1 immunoreactive primary afferents, 47 and 59 in the first and second groups, respectively. We found 9 and 19 PAX2 positive cells among the cells that were obtained from groups 1 and 2, respectively (Figure 10A–C). The cell bodies of the recovered neurons were primarily in laminae II and III (93 of the 106 neurons), and only 7 and 5 were found in laminae I and IV, respectively. Interestingly, however, neurons receiving close appositions from IB4-binding primary afferents were located slightly more dorsally than those that were contacted by VGLUT1 immunoreactive boutons. The laminar distribution of the somata of neurons making close appositions with IB4-binding boutons was as follows: 7 in lamina I, 23 in lamina II, and 17 in lamina III. However, the ones receiving close appositions from VGLUT1 immunoreactive axon terminals were found in slightly deeper layers: 4 in lamina II, 50 in lamina III, and 5 in lamina IV. The dendritic trees of all observed neurons were reconstructed from a series of confocal sections. Presumably due to the varying levels of tdTomato expression in the dendritic trees of the labeled neurons, the size of the reconstructed dendritic trees varied in a wide range, and the geometries of the dendritic trees were far from being homogeneous. Most of the neurons contacted by IB4-binding axon terminals showed a vertical/stalked cell morphology [68,69] (Figure 10D), whereas most of the neurons receiving VGLUT1 immunoreactive contacts could be classified as islet or central cells [69] (Figure 10E).

On the dendrites of the 47 neurons in the sections stained for GlyT2 and IB4-binding, there were 1520 and 1306 close appositions with GlyT2 immunoreactive and IB4-binding axon terminals, respectively. Combining these values with the total lengths of the dendrites, we found that there were 4.4 ± 2.9 and 4.8 ± 1.9 close appositions made by IB4-binding and GlyT2 immunoreactive axon terminals, respectively, per 100 µm long dendritic segment. On the dendrites of the 59 neurons in the sections stained for GlyT2 and VGLUT1, we counted 1763 and 1019 contacts made by GlyT2 immunoreactive and VGLUT1 immunoreactive axon terminals, respectively (Figure 10F–I). We calculated the densities of these contacts on dendrites and found 2.5 ± 1.3 and 4.5 ± 2.3 close appositions made by VGLUT1 immunoreactive and GlyT2 immunoreactive axon terminals, respectively, per 100 µm long dendritic segment. The close appositions, regardless of the markers, were homogeneously distributed along the dendrites, and they did not show any sign of clustering. We did not find any significant differences in these parameters between PAX-positive and PAX-negative neurons.

Finally, we studied contacts made by GlyT2 immunoreactive boutons on IB4-binding and VGLUT1 immunoreactive axon terminals making close appositions with tdTomato-labeled dendrites (Figure 10J). For 22 (6 with IB4-binding and 16 with VGLUT1 immunoreactive axon terminals) of the 106 reconstructed neurons, we did not find any GlyT2 immunoreactive axo-axonic appositions. Then, we analyzed only the neurons on which GlyT2 axon terminals were found in axo-axonic close appositions. On the 41 cells stained with the anti-GlyT2 + IB4-binding combination, we found that 101 (9.2%) of the 1095 IB4-binding axon terminals making close appositions with the labeled dendrites were contacted by GlyT2 immunoreactive boutons. On the 43 cells stained with the anti-GlyT2 + anti-VGLUT1 combination, we found 765 VGLUT1 immunoreactive axon terminals making close appositions with the labeled dendrites, and 126 (16.4%) of the 765 axon terminals were contacted by GlyT2 immunoreactive boutons.

## 3. Discussion

### 3.1. GABAergic versus Glycinergic Inhibition in Laminae I–III of the Spinal Dorsal Horn

Fast inhibitory transmission is mediated by GABA and glycine in neural circuits of the spinal dorsal horn. The relative contribution of GABAergic and glycinergic inhibition to spinal pain processing is, however, continuously under debate. The major points of this debate were clearly explicated in a recent paper [16] in which the authors investigated GABA_A_ and glycine receptor-mediated inhibition of PKCγ-containing interneurons in inner lamina II of the spinal and medullary dorsal horn with pharmacological and anatomical methods. They claimed that nearly all (91.7%) inhibitory axon terminals making contact with these neurons contained GAD and approximately half of them (42.2% of all boutons) also expressed GlyT2; thus, nearly all of them could be regarded as GABAergic axon terminals, and half of the GABAergic axon terminals could also be considered glycinergic. They found only a few boutons (6.3%) that were positive for GlyT2 but negative for GAD. Considering these findings, it is surprising that in another set of experiments, they demonstrated that both GABA_A_ and glycine receptors were anchored in the postsynaptic membranes of 78.3% of these inhibitory synapses, whereas GABA_A_ receptors without glycine receptors were found in only 2.3% of the investigated postsynaptic membranes. Even more remarkably, however, by recording sIPSCs and mIPSCs from the same set of synapses, it was found that approximately 70% of the quantal events were mediated by either glycine-only or GABA-only synaptic transmission. The major disagreement between morphological and physiological results, which was clearly presented in the article of El Khoueiry et al. [16], became obvious several years ago and made the interpretation of data regarding the role of GABAergic and glycinergic inhibition in spinal pain processing quite ambiguous [1,2,9,10,12,14,23].

Our present FISH results confirm some of these earlier findings. Namely, we found that one third (32.9%) of the PAX2 positive inhibitory interneurons in laminae I–III expressed only GAD65/67 mRNAs indicating that they can mediate only GABAergic synaptic inhibition. The concept of glycine-only synaptic events seems to be slightly more problematic because according to our results, two thirds (67.1%) of the PAX2 positive inhibitory interneurons in laminae I–III expressed both GlyT2 and GAD65/67 mRNA, with GAD65/67 showing clear dominance. Thus, it is likely that inhibitory axon terminals arising from neurons in laminae I–III can mediate GABA-only or mixed GABA-glycine synaptic events. However, in contrast to early [4,5,6] and recent [16] immunohistochemical findings, Miranda et al. [26] recently reported that there are abundant axon terminals that are positive for GlyT2 and negative for GAD65/67 in laminae I–III, confirming electrophysiological data suggesting that in lamina II, the glycinergic component of inhibitory postsynaptic currents is as strong as (if not stronger than) the GABAergic component [1,70]. It has also been suggested that these glycine-only axon terminals may arise from neurons with cell bodies in lamina IV [26,47]. Although we found that there were few neurons positive for only GlyT2 mRNA (3.7%) and that most neurons (89.8%) expressed both GlyT2 and GAD65/67 mRNA in lamina IV, we also demonstrated that most of these double-labeled neurons showed remarkable GlyT2 mRNA dominance. Thus, one may assume that the glycine-only axon terminals in laminae I–III arise from neurons in lamina IV that express only GlyT2 mRNA or express both GlyT2 and GAD65/67 mRNAs with a strong GlyT2 dominance. In cells with high GlyT2 mRNA and low GAD65/67 mRNA expression, the quantity of GAD65/67 transported to the axon terminals and therefore the concentration of GABA in the axoplasm can be very low, so GAD65/67 could not be detected by Miranda et al. (2022) with immunocytochemical methods, and the concentration of GABA in the axoplasm is likely below the appropriate level for vesicular GABA transporter (VIAAT) uptake [71,72]. Thus, because of a lack or minimal amount of GABA in synaptic vesicles, the postsynaptic effect of GABA in these synapses cannot be detected with physiological and pharmacological techniques. El Khoueiry et al. [16] presumably developed a supersensitive immunohistochemical method that can likely detect minimal amounts of peptides (e.g., GAD and GABA_A_ receptors) with limited, if any, functional significance. As a matter of course, these notions need further experimental verification. However, when pain-related synaptic events are evaluated in the spinal dorsal horn, the possible contribution of glycine-only and glycine-dominant neurons in lamina IV to spinal pain processing, in addition to that of neurons in laminae I–III, should not be neglected. This possibly strong glycinergic input to neurons in laminae I–III may have an additional functional implication. Glycine currents have fast kinetics with rapid decay [73], features thought to be important for precisely timed inhibition of locomotor circuits [74,75]. The need for such precise inhibitory control can equally be important in spinal pain processing and favors the concept of glycine helping to segregate functionally distinct afferent signals.

### 3.2. Synaptic Targets of Glycinergic Axon Terminals of Spinal Origin in Laminae I–III of the Spinal Dorsal Horn

As discussed in the previous section, we propose that glycine can be released alone and together with GABA from inhibitory axon terminals in laminae I–III of the spinal dorsal horn. However, we investigated GlyT2-containing axon terminals but did not study glycine-only and mixed glycine-GABA terminals separately. Thus, we cannot tell whether the GlyT2-containing axon terminals that we identified here can release glycine or glycine together with GABA. For this reason, one can identify them simply as glycinergic.

Glycinergic axon terminals in laminae I–III may arise from local spinal neurons or may represent terminals of axons descending from brainstem nuclei [27,28]. Here, we showed that axons descending from the brain stem substantially contribute to the glycinergic innervation of lamina I–III. According to our data, axon terminals of the brain stem origin represent 32–40% of glycinergic boutons in laminae I–III. Thus, if one wishes to study glycinergic axon terminals of spinal origin, those that arise from the brain stem must be eliminated. In the present work, we eliminated descending axons through low-thoracic hemisections; thus, we can state that only glycinergic terminals of spinal origin were investigated in our studies.

#### 3.2.1. Postsynaptic versus Presynaptic Inhibition

In agreement with early findings [76], we showed here that GlyT2 immunoreactive axon terminals formed axo-dendritic, axosomatic, and axo-axonic synapses, confirming that glycine can mediate both post- and presynaptic inhibition [37,38,77,78]. Our present results obtained from different experimental approaches showed that approximately 85–90% of GlyT2 immunostained axon terminals formed axo-dendritic and axosomatic synapses and close appositions. Thus, our results clearly indicate that glycine-mediated postsynaptic inhibition must be much more prominent than glycinergic presynaptic inhibition. In addition, our present results also suggest that glycinergic postsynaptic inhibition must be a common functional property of neural circuits in laminae I–III of the spinal dorsal horn, whereas the glycine-mediated presynaptic inhibition may affect only specific sets of axon terminals.

#### 3.2.2. Targets of Postsynaptic Glycinergic Inhibition

The importance of postsynaptic glycinergic inhibition in acute and chronic spinal pain processing has been substantiated by a great deal of experimental data [1,11,12,14,16,19,22,54,63,79]. Most of these studies have found that glycinergic postsynaptic inhibition affects various sets of excitatory interneurons, but reports about glycinergic innervation of inhibitory interneurons are sporadic [11,53,70,80]. Presumably, this is the reason why neural circuit models of spinal pain processing created to date do not include any representation of glycinergic disinhibition achieved by glycinergic inhibition of inhibitory interneurons. However, here, we showed that GlyT2 immunoreactive axon terminals established close somatic appositions with almost all types of inhibitory interneurons in laminae I–III of the spinal dorsal horn, including PV-, CR-, PKCγ-, nNOS- and GAL-containing cells [35,63]. Thus, glycinergic inhibition of inhibitory interneurons may be very important in spinal pain processing. Of particular importance, the glycinergic innervation of PV-containing inhibitory interneurons [80], should be taken into consideration when their role in the generation of allodynia is discussed.

#### 3.2.3. Targets of Presynaptic Glycinergic Inhibition

Although presynaptic glycinergic inhibition does not seem to be as common in laminae I–III of the spinal dorsal horn as postsynaptic inhibition, it can substantially inhibit specific sets of primary afferents. According to our present results, only IB4-binding and VGLUT1 immunoreactive axon terminals, representing nociceptive nonpeptidergic C and nonnociceptive myelinated Aβ primary afferents, respectively, can receive presynaptic innervation from glycinergic neurons. It has been shown that these presynaptic glycinergic axon terminals may arise from PV-containing inhibitory neurons [37,38,77,78], but our present results indicate that glycinergic neurons in lamina IV may also contribute to the presynaptic innervation of IB4-binding and VGLUT1 immunoreactive axon terminals in laminae II–III. It is interesting to note that the primary afferent and glycinergic innervation of PKCγ-containing neurons, playing a major role in the generation of mechanical allodynia, show remarkably unique features. PKCγ-containing neurons in laminae I–IIo receive innervation from IB4-binding nonpeptidergic nociceptive C fibers [34], whereas PKCγ-containing neurons in laminae IIi-III are innervated by VGLUT1 immunoreactive nonnociceptive myelinated Aβ fibers [12,37,61,62]. Interestingly, as we showed here, a proportion of both sets of primary afferent inputs making synaptic contacts with the two distinct groups of PKCγ-containing neurons are under presynaptic glycinergic inhibition.

In contrast, Gradwell et al. [43,80] reported that although PV-containing inhibitory interneurons (iPVINs) form axo-axonic presynaptic inhibitory synapses with nonnociceptive Aδ and Aβ myelinated afferent fibers, iPVINs mediate only postsynaptic glycinergic inhibition, whereas iPVIN-derived presynaptic inhibition is mediated only by GABA. This observation by Gradwell et al. [43], however, does not exclude the possible existence of presynaptic glycinergic inhibition. First, on one fifth of the investigated PKCγ-containing neurons, we did not find any presynaptic axo-axonic contacts made by GlyT2 immunoreactive terminals. Thus, glycinergic presynaptic inhibition may affect only specific sets of neurons, even among a population of neurons that can be identified by a well-defined neuronal marker. Second, most of the GlyT2 immunostained axon terminals (85–90%) made axo-dendritic and axosomatic contacts in our different experiments, and only the rest of the terminals were found to form axo-axonic appositions. Thus, glycinergic postsynaptic inhibition may strongly dominate over glycinergic presynaptic inhibition. This may reflect the possibility that most glycinergic neurons mediate only postsynaptic inhibition, and only a fraction of them can mediate both post- and presynaptic inhibition. Third, we observed axo-axonic contacts made by GlyT2 immunoreactive terminals on approximately one-tenth of IB4-binding and one sixth of VGLUT1 immunoreactive axon terminals of primary afferents making contacts with the tdTomato-labeled PKCγ-containing neurons. Thus, presynaptic glycinergic inhibition at the cellular level can be weak, if detectable at all. However, one can suggest that glycinergic presynaptic inhibition, although weak at the cellular level, may be important for locally targeting functionally specific subpopulations of primary afferent inputs.

## 4. Material and Methods

### 4.1. Animals

Experiments were performed on three groups of mice: (1) wild-type B6 mice, (2) Tg(Pax2-cre)1Akg/Mmnc mice (Stock No: 010569-UNC, Mutant Mouse Regional Resource Centers) expressing Cre recombinase under the transcriptional control of the Pax2 gene, and (3) B6 129S6-Prkcgtm2/cre/ERT2)Ddg/J mice (stock No: 030289, The Jackson Laboratory, Bar Harbor, ME, USA) expressing CreERT2 recombinase under the transcriptional control of the gamma isoform of protein kinase C (PKCγ) gene. Both Pax2cre and Prkcgtm2/cre/ERT2 mice were crossed with a tdTomato reporter mice (007914—B6.Cg-*Gt(ROSA)26Sor^tm14(CAG-tdTomato)Hze^*/J, The Jackson Laboratory, Bar Harbor, ME, USA) (Appendix A). The offspring of the Prkcgtm2/cre/ERT2—tdTomato animals received an intraperitoneal tamoxifen injection (2 mg/pup, Sigma, St. Louis, MO, USA, Cat# T5648) on the 8–10th postnatal day (Appendix A). Tamoxifen solution was prepared as described by Zheng et al. [81]. On the 18–20th postnatal day, the offspring of both the Pax2cre-tdTomato and Prkcgtm2/cre/ERT2-tdTomato crosses were genotyped for both the Pax2 and tdTomato, and the PrkcgERT2 and tdTomato transgenes, respectively, and only those animals that expressed both genes were kept for the experiments. These transgenic animals and wild-type B6 mice were used for experiments at the age of 3–5 months. All animal study protocols were approved by the Animal Care and Protection Committee at the University of Debrecen and were performed in accordance with the European Community Council Directives.

### 4.2. Hemisection of the Spinal Cord

Under deep sodium pentobarbital anesthesia (50 mg/kg, i.p.) all animals that were used for the experiments were subjected to a laminectomy at the level of the 9–10th thoracic vertebrae. One half of the spinal cord at the level of the 11–12th thoracic spinal segment was transected, keeping the other side intact. The back muscles and the skin were sutured layer by layer, and the animals were allowed to wake up from the anesthesia. Further experiments were performed after a three-four week-long survival period (histotechnical details are given below). To confirm the proper execution of the hemisection, tissue sections were prepared from the site of hemisection and stained with cresyl violet (Figure 1). Other experiments were performed on sections obtained from the hemisected side of the lumbar spinal cord at the level of L4–L5 segments.

### 4.3. Preparation of Tissue Sections

#### 4.3.1. For Immunohistochemistry

Animals were deeply anesthetized with sodium pentobarbital (50 mg/kg, i.p.) and transcardially perfused with Tyrode’s solution (oxygenated with a mixture of 95% O_2_, and 5% CO_2_), followed by a fixative containing 4% paraformaldehyde dissolved in 0.1 M phosphate buffer (pH 7.4) for light microscopy, or 2.5% paraformaldehyde and 0.5% glutaraldehyde dissolved in 0.15 M cacodylate buffer (pH 7.4) for electron microscopy. After transcardial perfusion, the lumbar segments of the spinal cord were removed, postfixed in the original fixative for 3–4 h, and immersed in 10% and 20% sucrose dissolved in 0.1 M phosphate buffer or 0.15 M cacodylate buffer until they sank. The removed spinal cord was freeze-thawed in liquid nitrogen three times to enhance reagent penetration, sectioned at 50 µm on a vibratome, and extensively washed in the buffer in which the fixative was dissolved. Sections that were fixed for electron microscopy were treated with 0.1% H_2_O_2_ and 1% sodium borohydride for 15 and 30 min, respectively.

#### 4.3.2. For In Situ Hybridization

Animals were deeply anesthetized with sodium pentobarbital (50 mg/kg, i.p.). The lumbar segments of the spinal cord were removed and frozen in liquid nitrogen. The frozen spinal cord was embedded in a cryo-embedding medium, cut into 16 µm thick sections on a cryostat, and mounted onto Superfrost Plus glass slides (Thermo Fischer Scientific, Waltham, MA, USA). The sections were stored at −20 °C until further treatment.

### 4.4. Immunohistochemistry for Light Microscopy

#### 4.4.1. Single Immunostaining of Sections Obtained from Wild-Type Animals

Single immunostaining for GlyT2 was performed to assess the effect of low-thoracic hemisection on GlyT2-containing axon terminals within the lumbar spinal cord. Free-floating sections fixed with 4% paraformaldehyde were incubated with guinea pig anti-glycine transporter 2 (GlyT2) (1:40,000, Synaptic Systems, Göttingen, Germany, Cat# 272 004, RRID:AB_2619998) for 2 days at 4 °C and transferred to goat anti-guinea pig IgG secondary antibody conjugated with Alexa Fluor 488 (diluted 1:1000, Thermo Fisher Scientific, Cat# A-11073, RRID:AB_2537117) overnight. Before the antibody treatments, the sections were kept in 20% normal goat serum (Vector Labs., Burlingame, CA, USA) for 50 min. The antibodies were diluted in 10 mM TPBS (pH 7.4) to which 1% normal goat serum (Vector Labs., Burlingame, CA, USA) was added. The sections were mounted on glass slides and covered with Vectashield mounting medium (Vector Labs., Burlingame, CA, USA). List of primary antibodies used is given in Table 3.

#### 4.4.2. Double Immunostaining of Sections Obtained from Wild-Type Animals

To reveal close appositions between GlyT2-IR axon terminals and the axon terminals of nociceptive and nonnociceptive primary afferents as well as intrinsic excitatory spinal neurons, free-floating sections fixed with 4% paraformaldehyde were incubated with a mixture of guinea pig anti-glycine transporter 2 (GlyT2) (1:40,000, Synaptic Systems Cat# 272 004, RRID:AB_2619998) and one of the following antibodies, which were used to label terminals of different types of axons: (1) rabbit anti-calcitonin gene-related peptide (CGRP, 1:3000, Peninsula Laboratories, Augst, Switzerland, Cat# T-4239; RRID:AB_518150), (2) biotinylated isolectin B4 (b-IB4, 1:2000, Thermo Fisher Scientific, Cat# I21414), (3) rabbit anti-vesicular glutamate transporter 2 (VGLUT2, 1:1000, ABCAM, Cambridge, UK, Cat# AB216463; RRID:AB_2893024), or (4) rabbit anti-vesicular glutamate transporter 1 (VGLUT1, 1:2000, ABCAM, Cat#, AB227805; RRID:AB_2868428). The sections were incubated in the primary antibody solutions for 2 days at 4 °C and were transferred to a mixture of goat anti-guinea pig IgG conjugated with Alexa Fluor 555 (diluted 1:1000, Thermo Fisher Scientific, Cat# A-21435, RRID:AB_2535856) and goat anti-rabbit IgG conjugated with Alexa Fluor 488 (diluted 1:1000, Thermo Fisher Scientific Cat# A11034, RRID:AB_2576217) or streptavidin conjugated with Alexa Flour 488 (1:1000, Thermo Fisher Scientific, Cat# S11223) overnight. Before the antibody treatments, the sections were kept in 20% normal goat serum (Vector Labs., Burlingame, CA, USA) for 50 min. The antibodies were diluted in 10 mM TPBS (pH 7.4) to which 1% normal goat serum (Vector Labs., Burlingame, CA, USA) was added. The sections were mounted on glass slides and covered with Vectashield mounting medium (Vector Labs., Burlingame, CA, USA).

#### 4.4.3. Double Immunostaining on Sections Obtained from PAX2:Cre-tdTomato Transgenic Animals

To reveal close appositions between GlyT2-IR axon terminals and the cell bodies of inhibitory and excitatory neurons expressing various cellular markers, we performed double immunostaining of sections obtained from PAX2:Cre-tdTomato transgenic animals. Free-floating sections fixed with 4% paraformaldehyde were incubated with a mixture of guinea pig anti-GlyT2 (1:40,000, Synaptic Systems Cat# 272 004, RRID:AB_2619998) and one of the following antibodies, which were used to label different cell types: (1) rabbit anti-neuronal nitric oxide synthase (nNOS) (diluted 1:4000, Abcam Cat# AB76067, RRID:AB_2152469), (2) rabbit anti-parvalbumin (PV) (diluted 1:60,000, Swant AG, Burgdorf, Switzerland, Cat# PV27, RRID: AB_2631173), (3) rabbit anti-galanin (GAL) (diluted 1:10,000, Peninsula Laboratories Cat# T-4334, RRID:AB_518348), (4) rabbit anti-calretinin (CR) (diluted 1:5000, Swant Cat# 7697, RRID:AB_2721226), (5) rabbit anti-calbindin (CaB) (diluted 1:10,000, Swant, Cat# CB38, RRID:AB_10000340, (6) rabbit anti-gamma isoform of protein kinase C (PKCγ) (diluted 1:2000, Abcam, Cat# AB71558, RRID:AB_1281066). The sections were incubated in the primary antibody solutions for 2 days at 4 °C and transferred to a mixture of goat anti-guinea pig conjugated with Alexa Fluor 647 (diluted 1:1000, Thermo Fisher Scientific, Cat# A21450, RRID:AB_2735091) and goat anti-rabbit IgG conjugated with Alexa Fluor 488 (diluted 1:1000, Thermo Fisher Scientific, Cat# A11034, RRID:AB_2535849) secondary antibodies overnight. Before the antibody treatments, the sections were kept in 20% normal goat serum (Vector Labs., Burlingame, CA, USA) for 50 min. The antibodies were diluted in 10 mM TPBS (pH 7.4) to which 1% normal goat serum (Vector Labs., Burlingame, CA, USA) was added. The sections were mounted on glass slides and covered with Vectashield mounting medium (Vector Labs., Burlingame, CA, USA).

#### 4.4.4. Triple Immunostaining on Sections Obtained from the PKCγ:Cre/ERT2-tdTomato Transgenic Animals

To study the distribution of GlyT2-IR axon terminals which can potentially exert pre- or postsynaptic inhibitory effects on the somatodendritic membrane of excitatory and inhibitory PKCγ-containing neurons, we performed triple immunostaining of sections obtained from PKCγ:Cre/ERT2-TtdTomato transgenic animals. Free-floating sections fixed with 4% paraformaldehyde were incubated with a mixture of guinea pig anti-GlyT2 (1:40,000, Synaptic Systems Cat# 272 004, RRID:AB_2619998), rabbit anti-PAX2 (diluted 1:200, Thermo Fisher Scientific, Cat#71-6000, RRID: AB_2533990) and either biotinylated IB4 (b-IB4, 1:2000, Thermo Fisher Scientific, Cat# I21414), or goat anti-vesicular glutamate transporter 1 (VGLUT1, 1:5000, Synaptic System, Cat# 135 307; RRID: AB_2619821). The sections were incubated in the primary antibody solutions for 2 days at 4 °C and transferred to a mixture of donkey anti-guinea pig IgG conjugated with Alexa Fluor 647 (diluted 1:1000, Jackson ImmunoResearch, West Grove, PA, USA, Cat# 706-605-148, RRID: AB_2340476), donkey anti-rabbit conjugated with 405 (diluted 1:1000, Thermo Fisher Scientific, Cat# A48258, RRID: AB_2890547) and donkey anti-goat IgG conjugated with Alexa Fluor 488 (diluted 1:1000, Thermo Fisher Scientific Cat# A11055, RRID: AB_2534102) or streptavidin conjugated with Alexa Flour 488 (1:1000, Thermo Fisher Scientific, Cat# S11223) overnight. Before the antibody treatments, the sections were kept in 20% normal goat serum (Vector Labs., Burlingame, CA, USA) for 50 min. The antibodies were diluted in 10 mM TPBS (pH 7.4) to which 1% normal goat serum (Vector Labs., Burlingame, CA, USA) was added. The sections were mounted on glass slides and covered with Vectashield mounting medium (Vector Labs., Burlingame, CA, USA).

### 4.5. Immunohistochemistry for Electron Microscopy

To visualize synapses formed by GlyT2-IR axon terminals, free-floating sections fixed with 2.5% paraformaldehyde and 0.5% glutaraldehyde were incubated with guinea pig anti-GlyT2 (1:40,000, Synaptic Systems Cat# 272 004, RRID: AB_2619998) for 2 days at 4 °C and transferred to biotinylated goat anti-guinea pig IgG secondary antibody (1:200, Vector Laboratories Cat# BA-7000, RRID: AB_2336132) overnight. Then, the sections were transferred to an avidin biotinylated horseradish peroxidase solution (ABC, 1:100, Vector laboratories, Cat# PK-4005) for 5 h, and the immunoreactivity was detected with nickel-intensified diaminobenzidine (NiDAB). The NiDAB end-product was further intensified with silver precipitation, which was stabilized with gold toning as described by Kalló et al. [82] and Bardóczi et al. [83]. The sections were then treated with a mixture of 0.5% OsO4 and 1.5% ferricyanide in 0.15 M cacodylate buffer for 15 min and dehydrated and embedded in Durcupan. Ultrathin (50–60 nm thick) sections were cut, collected onto Formwar-coated single-slot grids, and counterstained with 2% lead citrate. The sections were viewed with a JEOL1010 transmission electron microscope, and images were taken with an Olympus Veleta slow scan-cooled digital camera.

### 4.6. Multiplex Fluorescent In Situ Hybridization

To check the specificity of the tdTomato labeling in PAX2:Cre-tdTomato transgenic animals, an RNAScope^®^ Multiplex Fluorescent Assay (FISH, ACD, Bio-Techne, Minneapolis, MN, USA) was performed, according to the manufacturer’s pretreatment protocol for fresh frozen tissue (document no. 320513, rev. date to 11 May 2015) and detection protocol (document no. 320293-USM, rev. date 04092019). The following probes were used: Mm-GAD1 (GAD67, ACD, Bio-techne, Cat# 400951), Mm-GAD2 (GAD65, ACD, Bio-techne, Cat# 439371), Mm-Slc6a5-C2 (GlyT2, ACD, Bio-techne, Cat# 409741-C2), and tdTomato-C3 (ACD, Biotechne, Cat# 317041-C3). Sections were covered with Vectashield mounting medium containing DAPI (Vector Labs., Burlingame, CA, USA).

### 4.7. Confocal Microscopy

Series of 1 µm thick confocal optical sections with an 0.5 µm overlap were scanned with an Olympus FV3000 confocal microscope using a 40× oil-immersion lens (NA: 1.3). The confocal settings (laser power, confocal aperture, and gain) were identical for all sections in a given experiment, and care was taken to ensure that no pixels corresponding to puncta immunostained or hybridized for the markers were saturated. The scanned images were processed by Adobe Photoshop CS5 software. Most of the immunohistochemical staining was evaluated in 1 µm thick optical sections. In the case of FISH experiments, five consecutive confocal sections were merged, and thus we created 3 µm thick compressed optical sections, and the results were evaluated on these optical sections.

### 4.8. Neurolucida Reconstruction

The series of scanned images of sections taken from PKCγ:Cre/ERT2-TtdTomato transgenic animals and triple immunostained for GlyT2, PAX2, and VGLUT1 or IB4-binding were transferred to a Neurolucida system (Neurolucida v. 11.07, MicroBrightfied Bioscience, Williston, VT, USA). The cell bodies and dendritic trees of tdTomato-labeled PKCγ-containing neurons as well as GlyT2 and VGLUT1 immunostained or IB4-binding axon terminals making contact with tdTomato-labeled perikarya and dendrites were reconstructed from the image stacks.

### 4.9. Statistical Analysis

Changes in the number of GlyT2-immunostained boutons following hemisection of the spinal cord, and the colocalization of PAX2-tdTomato labeling with GAD and GlyT2 mRNAs as well as with various cellular markers were quantitatively analyzed. 

#### 4.9.1. Changes in the Numbers of GlyT2 Immunostained Boutons Following Hemisection of the Spinal Cord

We compared the numbers of GlyT2 immunostained boutons in laminae I–II and lamina III of L4–L5 the spinal cord in sections obtained from unoperated animals and animals subjected to hemisection of the Th11–Th12 spinal cord. For animals subjected to hemisection, immunostained boutons were counted ipsilateral to the hemisection. Because of the limited penetration of the anti-GlyT2 antibody into the sections, we obtained confocal images of the very superficial layer of the sections, not deeper than 2–3 µm from the surface. The immunostained puncta were counted manually according to the following method: a 10 × 10 standard square grid comprising squares with a length of 5 µm was placed onto the regions corresponding to laminae I–II and lamina III of the superficial spinal dorsal horn in the confocal images. The grid was placed on the images according to the following procedure: (a) The border between the dorsal column and the dorsal horn was easily identified based on the intensity of immunostaining. (b) The border between laminae II and III was approximated according to previous observations [84,85,86]. It has been repeatedly demonstrated in ultrastructural studies that there are almost no myelinated axons in lamina II, while there is an abundance of myelinated axons in lamina III. Thus, the border between laminae II and III can be defined quite precisely in ultrastructural studies, and the thickness of laminae I–II can be measured. (c) According to thorough cytoarchitectural studies, the thickness of lamina III is approximately the same as the thickness of laminae I and II together at the L4–5 level of the mouse spinal cord [87]. For this reason, immunolabeling was investigated in the most superficial 60 µm thick zone and another zone located 60–120 µm from the border between the dorsal column and the gray matter of the dorsal horn, which was previously found to correspond to laminae I–II and lamina III at the L4–L5 level of the mouse spinal dorsal horn [87].

Puncta immunostained for GlyT2 and located over the edges of the 5 µm unit squares were counted. The experiment was performed on three unoperated animals and three animals subjected to a hemisection. Five sections from each animal were randomly selected; thus, the quantitative data were collected from 15 independent sections.

#### 4.9.2. Colocalization of tdTomato, GAD1/2, and GlyT2 mRNAs

The colocalization of tdTomato, GAD1/2 (GAD67/65), and GlyT2 mRNAs was investigated in 3 µm thick optical sections obtained from PAX2:Cre-tdTomato transgenic animals and subjected to multiplex FISH. Neurons stained for the mRNAs of interest were counted in laminae I–II and lamina III. The borders of the laminae were defined as described in the previous section.

Neurons stained for mRNAs of interest were counted in three animals. Five sections from each animal were randomly selected; thus, the quantitative data were collected from 15 independent sections. 

#### 4.9.3. Expression of PAX2 in the Cell Bodies of Neurons Immunostained for Various Neuronal Markers

The colocalization of PAX2:Cre-tdTomato labeling with immunostaining for nNOS, GAL, PV, CaB, CR, and PKCγ was investigated as follows. Cell bodies labeled with the neuronal markers and positive or negative for PAX2:Cre-tdTomato were identified and counted in laminae I–III. The percentages of tdTomato-positive and tdTomato-negative neurons expressing each marker were calculated. Immunostained neurons were counted in three animals. Five sections from each animal were randomly selected; thus, the quantitative data were collected from 15 independent sections.

## Figures and Tables

**Figure 1 ijms-24-06943-f001:**
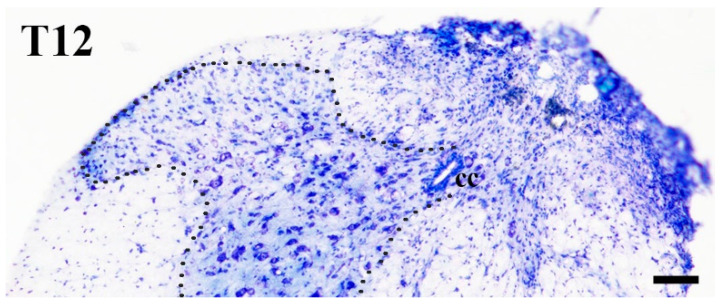
Hemisection of the spinal cord. Photomicrograph showing a section of the spinal cord obtained from the hemisected level and stained with cresyl violet. Note that one half of the spinal cord is intact, whereas due to hemisection, the other side does not show the normal cytoarchitectonic pattern of the spinal cord. Hemisection was performed at the Th12 level of the spinal cord. The dotted line on the intact side indicates the border between the gray and white matter. cc: central canal. Bar: 100 µm.

**Figure 2 ijms-24-06943-f002:**
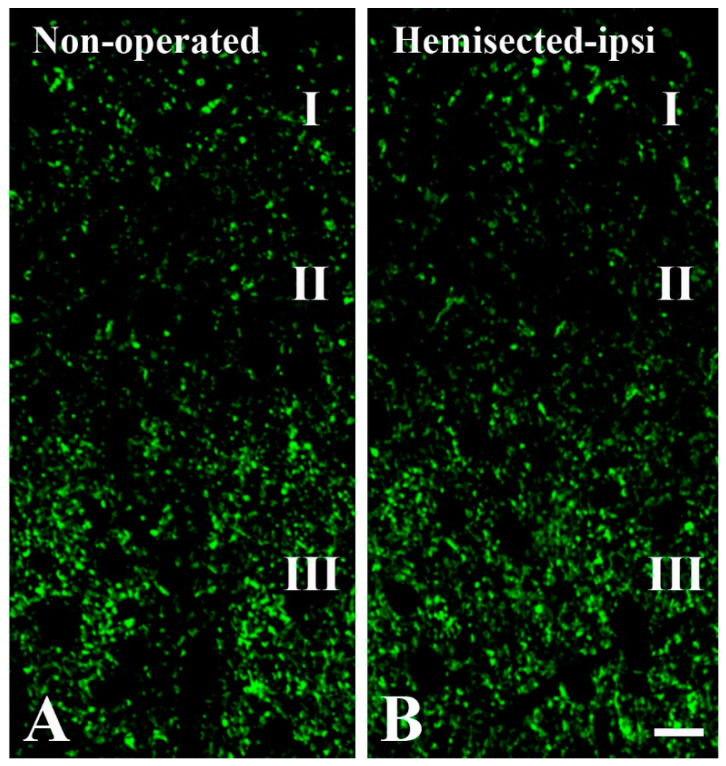
Effect of low thoracic hemisection on GlyT2 immunoreactivity of the lumbar spinal dorsal horn. Photomicrographs showing 1 µm thick optical sections of the ipsilateral superficial spinal dorsal horn at the L4 level from a nonoperated animal (**A**) and an animal subjected to hemisection (**B**), following immunostaining for GlyT2. Roman numerals indicate the layers (laminae I, II, and III) of the superficial spinal dorsal horn. Bar: 10 µm.

**Figure 3 ijms-24-06943-f003:**
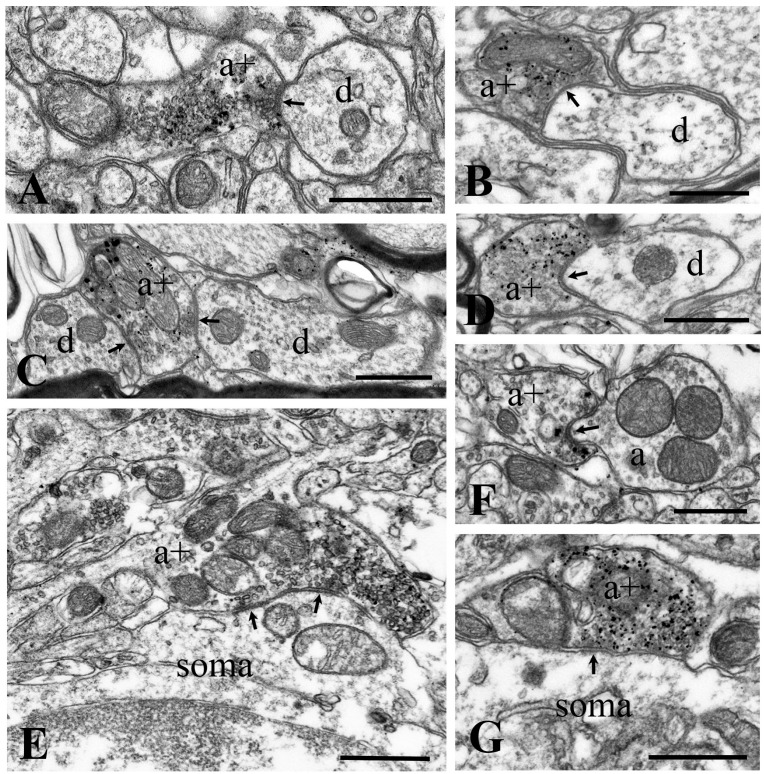
Synapses made by GlyT2 immunostained axon terminals. Electron micrographs showing axo-dendritic (**A**–**D**), axosomatic (**E**,**G**), and axo-axonic (**F**) synaptic contacts made by GlyT2 immunostained axon terminals in laminae I–III of the superficial spinal dorsal horn at the L4–L5 level of the spinal cord. GlyT2 immunoreactivity appears as accumulated gold particles at perisynaptic locations and is distributed over synaptic vesicles. a+: axon terminals immunostained for GlyT2, d: postsynaptic dendrites, soma: postsynaptic perikarya, and a: postsynaptic axon. Arrows point to synaptic appositions. Bars: 0.5 µm.

**Figure 4 ijms-24-06943-f004:**
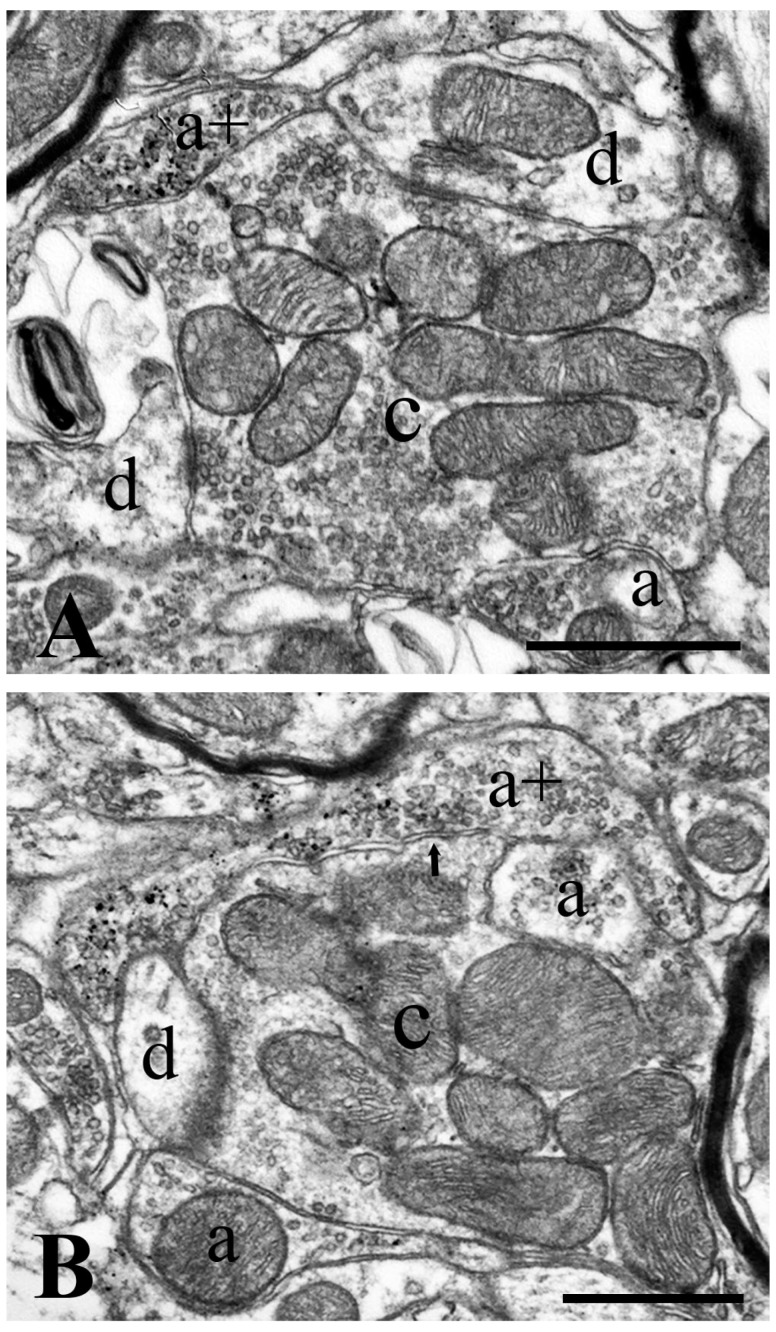
GlyT2 immunostained axon terminals in glomerular synaptic complexes. Electron micrographs showing glomerular synaptic complexes in laminae II–III of the superficial spinal dorsal horn at the L4–L5 levels of the spinal cord. Note that GlyT2 immunostained axon terminals make synaptic contacts with the central axons of the glomeruli. GlyT2 immunoreactivity appears as accumulated gold particles at perisynaptic locations and is distributed over synaptic vesicles (**A**,**B**). c: central axon terminal of the glomerular synaptic complex, a+: axon terminals immunostained for GlyT2, a: axon terminals negative for GlyT2, and d: dendrites postsynaptic to the central axon terminal. Arrows point to synaptic appositions. Bars: 0.5 µm.

**Figure 5 ijms-24-06943-f005:**
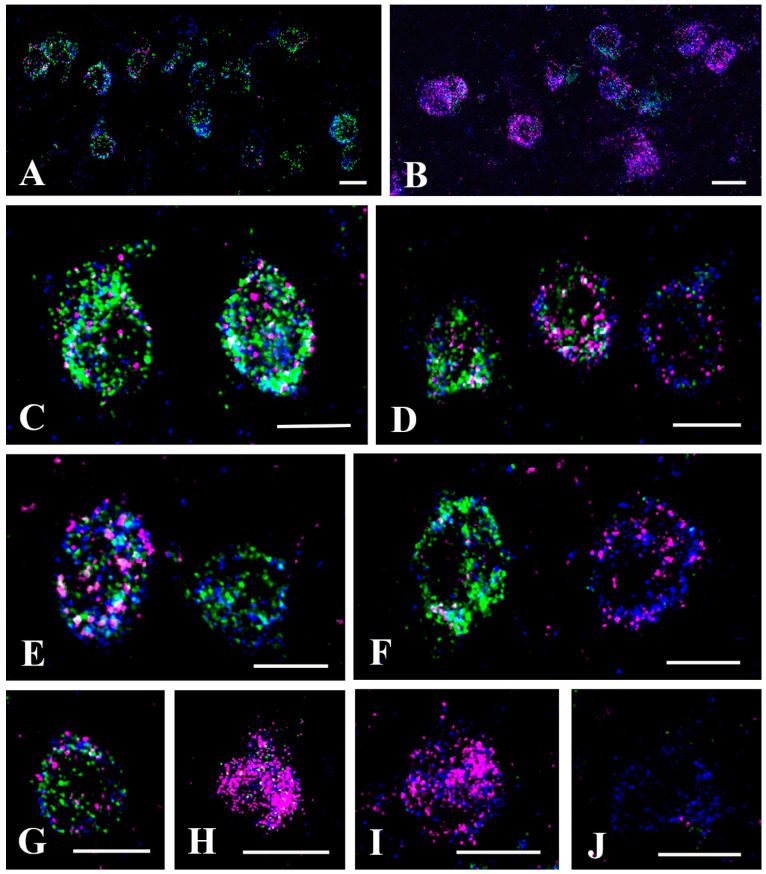
Expression of tdTomatomo, GlyT2, and GAD65/67 mRNA in the lumbar spinal dorsal horn of Pax2:Cre-tdTomato mice. Micrograph of 3 µm thick confocal optical sections obtained from the L4–5 levels of the spinal dorsal horn of Pax2:Cre-tdTomato mice and hybridized for tdTomato (blue), GlyT2 (magenta) and GAD65/67 (green) mRNA. The hybridization end products appear as small dots within the confines of individual neurons. Triple labeling images are shown at low (**A**,**B**) and high (**C**–**J**) magnifications. The images are of laminae I–III (**A**,**C**–**E**,**G**,**J**) and lamina IV (**B**,**F**,**H**,**I**) of the dorsal horn. Not that most of the neurons are triple labeled, but there are some that are negative for GlyT2 (**E**) or GAD 65/67 (**F**,**I**) or both GlyT2 and GAD 65/67 (**J**) mRNA. Bars: 20 µm.

**Figure 6 ijms-24-06943-f006:**
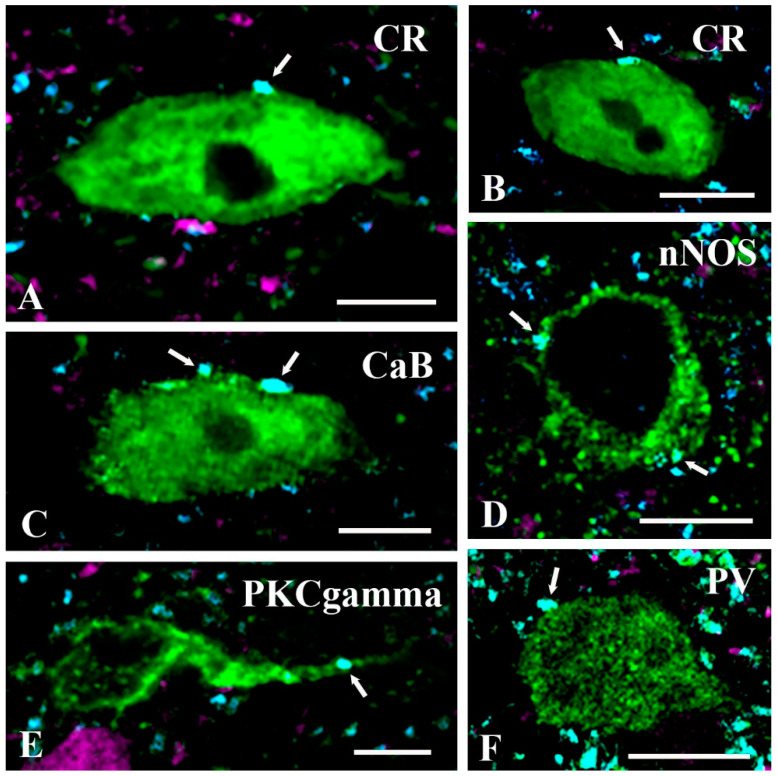
GlyT2 immunostained axon terminals make close appositions with excitatory spinal interneurons. Close appositions between glycine transporter 2 (GlyT2) immunoreactive axon terminals (cyan) and somata or proximal dendrites of calretinin (CR) (**A**,**B**), calbindin CaB) (**C**), neuronal nitric oxide synthase (nNOS) (**D**), protein kinase C gamma (PKCgamma) (**E**), and parvalbumin (PV) (**F**) immunoreactive (green) and PAX2Cre-tdTomato (magenta) negative excitatory interneurons. Arrows indicate GlyT2 immunostained axon terminals forming close appositions with the labeled neurons. Bars: 10 µm.

**Figure 7 ijms-24-06943-f007:**
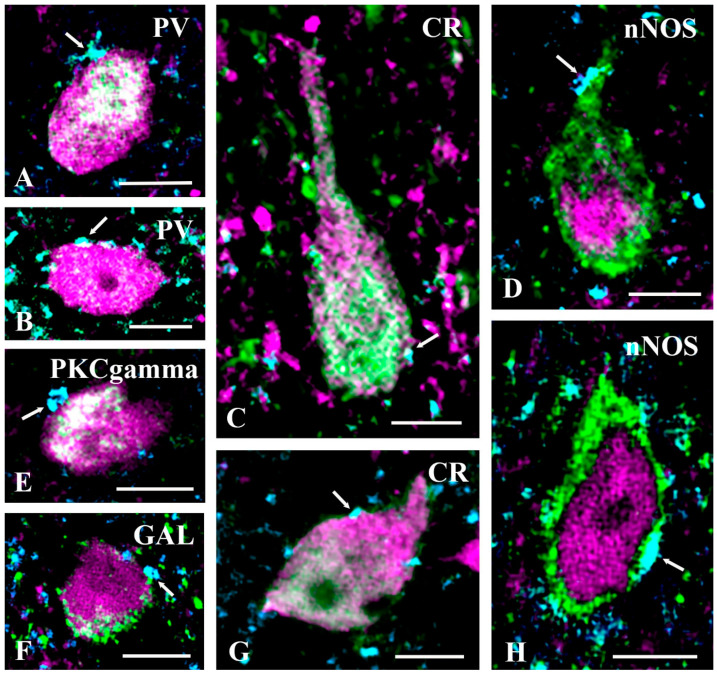
GlyT2 immunostained axon terminals make close appositions with inhibitory spinal interneurons. Close appositions between glycine transporter 2 (GlyT2) immunoreactive axon terminals (cyan) and the somata or proximal dendrites of parvalbumin (PV) (**A**,**B**), calretinin (CR) (**C**,**G**), neuronal nitric oxide synthase (nNOS) (**D**,**H**), protein kinase C gamma (PKCgamma) (**E**), and galanin (GAL) (**F**) (green) immunoreactive and PAX2Cre-tdTomato (magenta) positive inhibitory interneurons. Note the mixed colors of the cell bodies of the illustrated neurons. It is interesting to note that unlike in cells positive for the other neuronal markers, in the nNOS-positive neurons, nNOS immunostaining is restricted to the peripheral portion of the cytoplasm, whereas tdTomato can be observed in the central part of the cytoplasm. Arrows indicate GlyT2 immunostained axon terminals forming close appositions with the labeled neurons. Bars: 10 µm.

**Figure 8 ijms-24-06943-f008:**
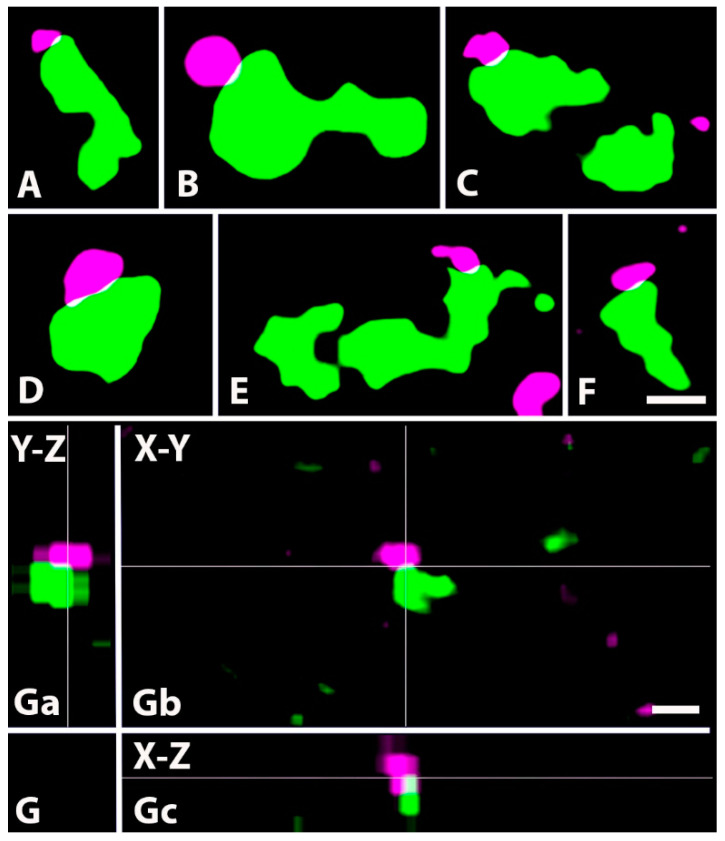
Close appositions between axon terminals immunoreactive for GlyT2 and VGLUT1. (**A**–**F**). Micrographs of 1 µm thick laser scanning confocal optical sections showing close appositions between axons immunoreactive for GlyT2 (magenta) and VGLUT1 (green), a marker for nonnociceptive primary afferents. Note the thin contact zones (yellow) between the two types of labeled axon terminals, which may represent axo-axonic synapses observed with electron microscopy. (**G**). Micrographs of a short series of confocal optical sections double immunostained for GlyT2 (magenta) and VGLUT1 (green) showing contact between GlyT2 (magenta) and VGLUT1 (green) immunostained axon terminals shown in X-Y (Gb), X-Z (Ga) and Y-Z (Gc) projections. The contact zone between the two labeled axon terminals (yellow) is at the crossing point of two lines indicating the planes through which orthogonal views of the X-Z and Y-Z projections were drawn. Note that the contact zone between the two labeled axon terminals (yellow) can be identified in all three orthogonal images. Scale bars: 0.5 µm (**A**–**F**) and 1 µm (**G**).

**Figure 9 ijms-24-06943-f009:**
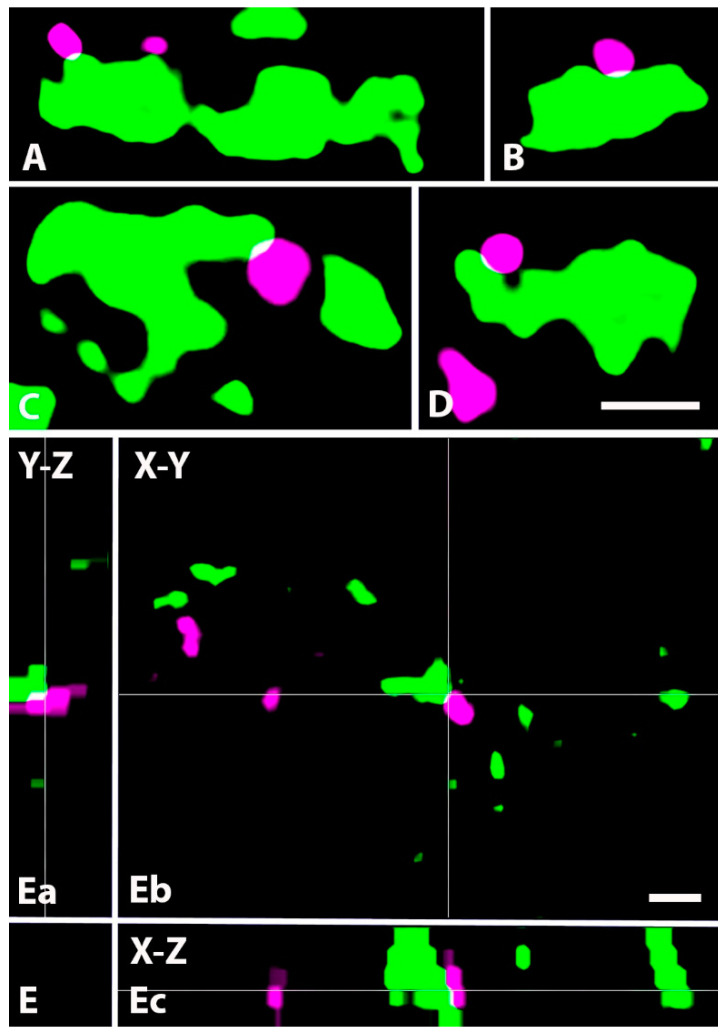
Close appositions between axon terminals immunoreactive for GlyT2 and binding IB4. (**A**–**D**). Micrographs of 1 µm thick laser scanning confocal optical sections showing close appositions between axon terminals immunoreactive for GlyT2 (magenta) and binding IB4 (green), a marker for nonpeptidergic nociceptive primary afferents. Note the thin contact zone (yellow) between the two types of labeled axon terminals, which may represent axo-axonic synapses observed with electron microscopy. (**E**). Micrographs of a short series of confocal optical sections double labeled for GlyT2 (magenta) and IB4-binding (green) showing contact between GlyT2 immunoreactive (magenta) and IB4 binding (green) axon terminals shown in X-Y (Eb), X-Z (Ea) and Y-Z (Ec) projections. The contact zone between the two labeled axon terminals is at the crossing point of two lines indicating the planes through which orthogonal views of X-Z and Y-Z projections were drawn. Note that the contact zone between the two labeled axon terminals (yellow) can be identified in all three orthogonal images. Scale bars: 0.5 µm (**A**–**D**), 1 µm (**E**).

**Figure 10 ijms-24-06943-f010:**
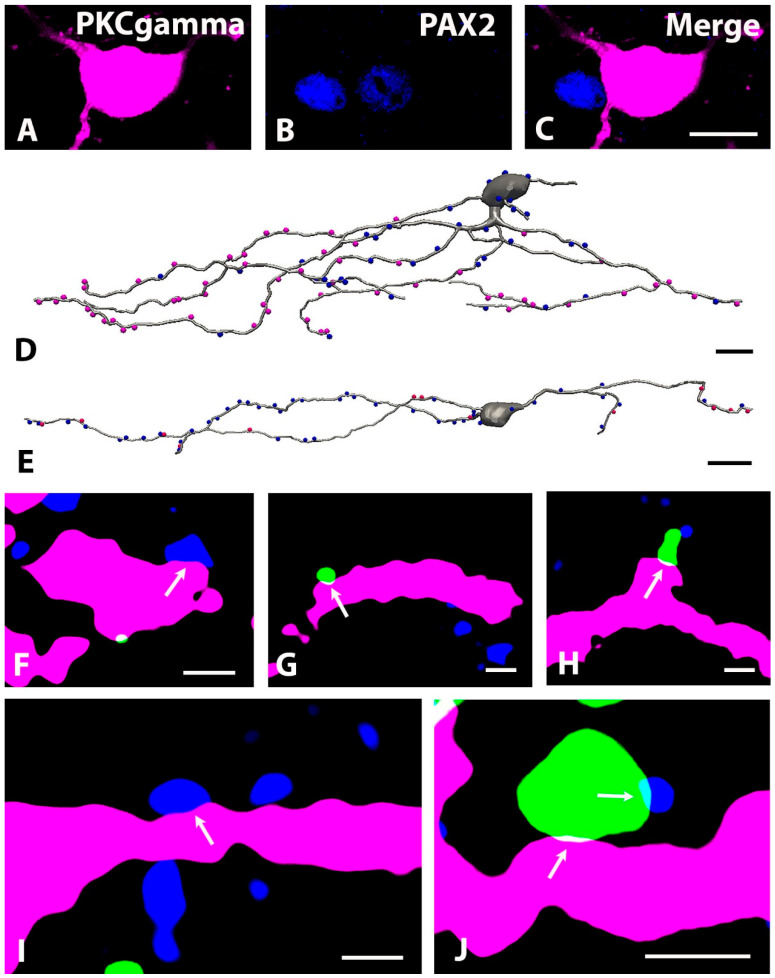
Close appositions between unmyelinated (IB4-binding) and myelinated (VGLUT1 immunostained) primary afferents as well as GlyT2 immunoreactive axon terminals on tdTomato-labeled neurons in Prkcgtm2/cre/ERT2-tdTomato mice. (**A**–**C**): Micrographs of a single 1 µm thick laser scanning confocal optical section showing PAX2 (blue) immunostaining of the nucleus of a tdTomato (magenta) labeled PKCγ-containing neuron. (**D**–**E**): Camera lucida reconstruction of the dendritic trees of PKCγ (gray)-containing neurons making close appositions with GlyT2 (blue) immunoreactive as well as IB4-binding (**D**) and VGLUT1 (**E**) immunostained axon terminals (magenta). (**F**–**J**): Micrographs of 1 µm thick laser scanning confocal optical sections showing close apposition between tdTomato (magenta) labeled dendrites of PKCγ-containing neurons and VGLUT1 (green; (**G**,**H**,**J**)) or GlyT2 (blue, (**F**,**I**)) immunoreactive axon terminals. Note that the VGLUT1 immunoreactive axon terminal forms a close apposition with a tdTomato-labeled dendrite in inset (**J**) contacted by a GlyT2 immunostained axon terminal. Arrows indicate the sites of close appositions between the labeled structures. Bars: 10 µm (**A**–**E**),1 µm (**J**) 0.5 µm (**F**–**I**).

**Table 1 ijms-24-06943-t001:** Numbers of neurons in which different combinations of tdTomato, GlyT2, and GAD65/67 mRNA were detected in laminae I–III and lamina IV of the spinal dorsal horn.

	1	2	3	4	5	6	7	8
	tdTomato+ GlyT2+ GAD+	tdTomato+ GlyT2+ GAD−	tdTomato+ GlyT2− GAD+	Ʃ1–3	tdTomato+ GlyT2− GAD−	Ʃ4–5	tdTomato− GlyT2+ and/or GAD+	Ʃ6–7
Laminae I–III	261	0	128	389	19	408	9	417
Lamina IV	589	24	43	656	9	665	2	667

**Table 2 ijms-24-06943-t002:** Numbers of neurons immunostained for different cellular markers and positive or negative for tdTomato (PAX2) in laminae I–III of the spinal dorsal horn.

Marker	CaB		PKCγ		CR		GAL		nNOS		PV	
	No.	%	No.	%	No.	%	No.	%	No.	%	No.	%
marker+/tdTomato+	11	3.1	32	13.2	62	17.9	19	90.5	36	58.1	87	45.3
marker+/tdTomato−	345	96.9	210	86.8	284	82.1	2	9.5	26	41.9	105	54.7
total	356	100	242	100	346	100	21	100	62	100	192	11

**Table 3 ijms-24-06943-t003:** List of primary antibodies used.

Target	Host Species	Dilution	Catalog #	Company	RRID
GlyT2	Guinea pig	1:40,000	272-004	Synaptic Systems	AB_2619998
Calbindin	Rabbit	1:10,000	CB38	SWANT	AB_10000340
Calretinin	Rabbit	1:5000	7697	SWANT	AB_2721226
CGRP	Rabbit	1:3000	T-4239	Peninsula	AB_518150
Galanin	Rabbit	1:10,000	T-4334	Peninsula	AB_518348
nNOS	Rabbit	1:4000	AB76067	Abcam	AB_2152469
Parvalbumin	Rabbit	1:60,000	PV27	SWANT	AB_2631173
PAX2	Rabbit	1:200	71-6000	ThermoFisher Scientific	AB_2533990
PKC-γ	Rabbit	1:2000	AB71558	Abcam	AB_1281066
VGLUT1	Rabbit	1:2000	AB227805	Abcam	AB_2868428
VGLUT1	Goat	1:5000	135 307	Synaptic Systems	AB_2619821
VGLUT2	Rabbit	1:1000	AB216463	Abcam	AB_2893024
b-IB4	-	1:2000	I21414	ThermoFisher Scientific	

## Data Availability

The data presented in this study are available on request from the corresponding author.

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
