# Peer review of "Synaptic Targets of Glycinergic Neurons in Laminae I–III of the Spinal Dorsal Horn"

_ijms, 2023, doi:10.3390/ijms24086943_

Round 1
Reviewer 1 Report
This is a potentially interesting article- the scientific rationale is sound and well presented. The flow is good, and builds a clear and succinct story (which is of interest to the field).
The mouse work is very interesting- it would be beneficial to include a summary figure at the start of the results section) that explains the crosses used and the differences between the mice studied (with justifications) at the start of the results section.
However, there are issues with data presentation and a lack of any statistical analysis, which means the paper reads as anecdotal and lacks validity. The experiments and data is presented as visual images, with count data- suggesting multiple counts on multiple sections for several figures / tables have been combined and only the top line data presented- this reduces the impact because it is not obvious the reader if these are true representations of the reported values. The authors should repeat counts on multiple sections, present the mean and statistics associated with each section and perform basic statistical analysis (t- or u-test data, with the SD) to demonstrate the presented data is truly representative and is demonstrating a confirmed biological variation.
Reviewer 2 Report
The present study uses a range of IHC, ISH, and advanced microscopy to further explore the synaptic targets of glycinergic axon terminals arising from glycinergic neurons in the spinal cord. To effectively isolate these inputs from descending inhibitory signals from brainstem or higher, most experiments are performed after hemisecting the spinal cord at Th11-12. Notably this procedure suggested ~40% of GlyT2 terminals are from descending axons, while ~60% are likely to be from glycinergic neurons whose soma is located in the spinal cord.
In hemisected animals, detailed anatomical and immunohistochemical studies are performed to determine the extent to which remaining glycine transporter 2 (GlyT2) positive axon terminals make axosomatic vs. axoaxonic contact with neurons or terminals in lamina I-III. To oversimplify a bit, the primary take home message is that GlyT2 axon terminals presumed to originate from spinal glycinergic neurons target nearly all types of excitatory and inhibitory cells examined in lamina I-III, but make axoaxonic connections to only a subset of terminals, including those nociceptive and non-nociceptive fibers. More broadly, about 85-90% of the GlyT2 positive terminals examined were making axosomatic or axodendritic contacts, while only 13% of GlyT2 positive terminals were making axoaxonic contacts. When it comes to predicting the extent to which inhibition from terminals examined is glycinergic vs. GABAergic, things get a little murkier, mostly because the strong majority of spinal GlyT2 positive neurons are also GAD65/67 positive. Broadly, the authors suggest that relative dominance of GlyT2 vs. GAD65/67 mRNA in a neuron may be predictive of extent to which glycinergic vs. GABAergic vs. mixed synaptic transmission is from axon terminals of that neuron likely. If this were the case, then the authors suggest synapses from GlyT2 neurons whose soma is in laminae I-III might trend towards predominantly GABAergic, and suggest glycinergic only synapses (which have been shown in other studies to exist) may be more likely to arise from mixed GlyT2/GAD positive neurons whose soma is in laminae IV (where GlyT2 expression has higher probability within a cell of being dominant).
In a broad sense, I have very few suggestions for revision or major comments for the authors to consider. The work is detailed, careful, and extensive. The writing is clear, the data is well illustrated, and the interpretations are in my opinion bot appropriate and appropriately cautious. While the paper does not really manage to definitively address core questions in the field about role and nature of glycinergic vs. GABAergic inhibitory transmission in the dorsal horn (either generally or as it relates to nociception), it does provide some new insight on axosomatic/dendritic vs. axoaxonic inhibitory synaptic contacts made by spinal GlyT2 positive neurons in laminae I-III, and provides some interesting speculation on what the findings may mean about physiological origin and relevance of predominately glycinergic transmission in places where that probably occurs. These advances represent a solid if incremental step. More broadly, it seems the technological advances necessary to address some of these long-standing questions with more definitive approaches that include not just anatomical but also physiological and functional measurements are becoming within reach, and it will be interesting to see what these and other authors do in this space in the future.
Reviewer 3 Report
The manuscript examined the spinal glycinergic synapse in the pain processing region of the spinal dorsal horn in mice by morphological approach. They used several neuronal markers to identify specific types or functions of neurons and used multiple labeling methods to clarify the synaptic targets of glycine neurons in the spinal dorsal horn. The manuscript is generally well-written, and the morphological results provided some interesting findings concerning the possible role of glycine synapse in spinal pain transmission. I have minor suggestions for the manuscript as follows:
- The thickness of the sections mentioned in the methods and results (legend) was inconsistent. For example, the thickness of the section is 16 um (for in situ hybridization, line 632); however, the thickness of the section is 3 um in figure 5.
- The results were abundant and complicated; a summary or conclusion of their findings might be required in the Discussion chapter.
- To render the reader understanding of the experimental design, it would be better to summarize the purpose of the makers used.
Round 2
Reviewer 1 Report
The authors have addressed each of my issues. I am happy for the paper to be accepted in its present form.